

# Predicting the salt water intrusion in the Shatt al Arab estuary using an analytical approach

Ali D. Abdullah,[1,2,3] * Jacqueline I. A. Gisen,[4,5] Pieter van der Zaag,[1,2] Hubert H.G. Savenije,[2] Usama F.A. Karim,[6] Ilyas Masih,[1] Ioana Popescu [1]

1. Department of Integrated Water System and Governance, UNESCO-IHE Institute for Water Education, Westvest 7, 2611 AX, Delft, the Netherlands
2. Water Resources Section, Delft University of Technology, Delft, the Netherlands
3. Department of Civil Engineering, Basra University, Basra, Iraq.
4. Centre for Earth Resources Research and Management, University Malaysia Pahang, Lebuhraya Tun Razak, Gambang, 26300 Kuantan, Pahang, Malaysia.
5. Faculty of Civil Engineering and Earth Resources, University Malaysia Pahang, Lebuhraya Tun Razak, Gambang, 26300 Kuantan, Pahang, Malaysia.
6. Department of Civil Engineering, University of Twente, P.O. Box 217, 7500 AE, Enschede, the Netherlands.
* Correspondence to: Ali D. Abdullah (a.abdullah@unesco-ihe.org and alidinar77@gmail.com)

**Abstract**

Longitudinal and vertical salinity measurements are used in this study to predict the extent of in-land seawater intrusion in a deltaic river estuary. A predictive model is constructed to apply to the specific tidal, seasonal and discharge variability and geometric characteristics of the Shatt al-Arab River (SAR) situated along the border of Iraq and Iran. Reliable hydrologic simulation of salinity dynamics and seawater intrusion was lacking prior to this study. Tidal excursion is simulated analytically using a 1-D analytical salt intrusion model with recently updated equations for tidal mixing. The model was applied under different river conditions to analyze the seasonal variability of salinity distribution during wet and dry periods near spring and neap tides between March 2014 and January 2015. A good fit is possible with this model between computed and observed salinity distribution. Estimating water abstractions along the estuary improves the performance of the equations, especially at low flows and with a well calibrated dispersion-excursion relationship of the updated equations. Salt intrusion lengths given the current data varied from 38 to 65 km during the year of observation. With extremely low river discharge, which is highly likely there, we predict a much further distance of 92 km. These new predictions demonstrate that the SAR, already plagued with extreme salinity, may face deteriorating water quality levels in the near future, requiring prompt interventions.

**Keywords:** Analytical model; Salt intrusion; Shatt al-Arab River; Alluvial estuaries





# 1 Introduction

Discharge of fresh river water into the ocean is closely related with vertical and longitudinal salinity variations along an estuary (e.g. Savenije et al, 2013; Whitney, 2010; Becker et al., 2010; Wong, 1995; MacKay and Schumann, 1990). River discharge also has a noticeable effect on the tidal range primarily through the friction term (the amount of energy per unit width lost by friction) (Savenije, 2005). A decrease of river discharge into an estuary could increase the tidal range and the wave celerity, and consequent increase in salinity levels (Cai et al., 2012). Upstream developments of large dams and water storage facilities change the nature of river flow and subsequently alter river hydrology and quality (Vorosmarty and Sahagian, 2000; Helland-Hansen et al., 1995). The Shatt al-Arab River (SAR) which discharges through its estuary at the border between Iran and Iraq into the Gulf is facing serious reductions in freshwater inflows upstream and from its tributaries, as well as significant salt intrusion downstream (Abdullah et al., 2015). The alteration of river discharge also affects the estuarine ecosystem in terms of sediments, nutrients, dissolved oxygen, and bottom topography (Sklar and Browder, 1998). All these problems are strongly featured in the SAR.

The increases in salinity along the SAR, particularly caused by salt intrusion, have become a threat to the people and environment alike. Generally, salt water intrusion makes the river water unfit for human consumption and unacceptable for irrigation practices (Abdullah et al., submitted; Al-Tawash et al., 2013). Saline water in the SAR estuary comes from both natural (sea water intrusion) and anthropogenic sources. Thus, the pattern of the salinity variation is complex because of the dynamic spatial and temporal interaction between salinity sources. Available studies on the SAR identify the escalating pressure of salinity increment and its consequences on water users as well as the ecosystem (e.g. Abdullah et al., submitted; Al-Tawash et al., 2013; Fawzi and Mahdi, 2014), but detailed information on the extent of salt water intrusion under different conditions is lacking. Hence, there is a need to investigate the impact of seawater intrusion among other sources on the river salinity, and to analyze the dynamics of the saline-fresh water interface for effective water management.

Different approaches have been used to study the relationship between saline and fresh water in estuaries. Alber (2002) proposed a conceptual model for managing freshwater discharge into estuaries. Wang et al. (2011) used an empirical approach, conducting three hydrological surveys along six locations around the Yellow River mouth to investigate the effect of abrupt changes in the river discharge on the salinity variations. Using a numerical model, Bobba (2002) analyzed the mechanism of salt water and freshwater flow in the Godavari Delta and found that freshwater withdrawals contribute to the advance in seawater intrusion. Lui et al. (2004) applied a two-dimensional model to estimate the salinity changes in the Tanshui River, showing that the significant salinity increase is a result of reservoir construction and bathymetric changes. A 3D model was used by Vaz et al. (2009) to study the patterns of saline water in the Espinheiro tidal channel. The result indicates that the model underestimated the salinity distributions for high river inflow. Das et al. (2012) used a hydrology-hydrodynamics model to examine salinity variations under different water diversion scenarios in the Barataria estuary, and discovered that the diversions have a strong impact on salinity in the middle section of the estuary and minor impact in the upper section.





Analytical approaches describing salinity distribution in estuaries have been used by Ippen and Harlemen (1961), Prandle (1985) and
Savenije (1986). An analytical solution is able to provide important knowledge about the relationship between tide, river flow, and
geometry of the tidal channel. The one-dimensional modelling is usually based on a number of assumptions to simplify the set of equations.
Several available models generally assumed a constant tidal channel cross-section to linearize and simplify the calculation processes. In
this study the 1-D analytical salt intrusion model proposed by Savenije (1986, 1989, and 1993) is considered, which uses the more natural
exponential geometry and requires a minimal amount of data. The model has been successfully applied to several single-channel estuaries
worldwide (e.g. Risley et al., 1993; Horrevoets et al., 2004; Gisen et al., 2015a). Moreover, it can also describe the tidal propagation in
multi-channel estuaries (Zhang et al, 2012) as well as estuaries with a slightly sloping bottom (Nguyen and Savenije, 2006; Cai et al.,
9  2015).
The aim of this study is to determine the real extent of salt intrusion into the SAR estuary. This is by applying the 1-D analytical salt
intrusion model combined with the revised predictive equations for tidal mixing of Gisen et al. (2015b). Then the predictive model was
used to examine the consequences of changes in river flow on the salinity distribution.
**2  Research area**
The SAR is located in southern Iraq and its estuary is connected to the Gulf (Fig. 1). The total length of the river is 195 km, of which the
last 95 km serves as a boundary between Iraq and Iran. The estuary receives fresh water from four main tributaries. The Tigris and
Euphrates rivers originate in Turkey and form the SAR at their confluence near the city of Qurna, Iraq. The other two tributaries, Karkheh
and Karun, originate in Iran. The Karkheh is connected with the SAR through a system of marshes, while the Karun discharges into the
SAR at approximately 87 km from the mouth.
The estuary experiences a tidal cycle of approximately 12 hours 25 minutes with notable flood and ebb tides (Fig. 2). The estuary has a
mixed-diurnal and semi-diurnal tide with successive spring and neap tide. The tidal range (the difference between the water levels at high
water (HW) and low water (LW)) varies from 1 m (neap) to 3 m (spring). Salinity levels fluctuate at an hourly scale depending on the tide
cycles and fresh water discharge. Salinity increases during flood tides and decreases during ebb. The impact of fresh water inflows can be
clearly recognized during neap tide and ebb periods. The salinity level also varies along the year, for example the highest value measured in
the year 2014 was 40 kg/m$^3$ during summer and the lowest value was 0.7 kg/m$^3$.
The SAR is the main surface water source for daily consumption and agricultural uses in the region and serves around 3 million people, the
majority living in Basra city. Rural communities live along the river and around the marshes and derive their livelihoods mainly from
agriculture and livestock. The main agricultural lands extend along the river banks with large date palm plantations. A variety of human





activities along the SAR and its tributaries deteriorates the water quality and has significantly increased the salinity concentration over time. In addition, the decreases of freshwater inflows into the estuary due to upstream water withdrawals have allowed the seawater to intrude further upstream. Currently the Tigris is the main source of fresh water for the SAR, its discharge ranges between 30- 100 $m^3$/s. The total discharge from other tributaries, except the Karun, ranges  between 0- 10 $m^3$/s. The available information on discharge of the Karun is limited and inconclusive. Most relevant is Ahvaz station in Iran (UN-ESCWA and BGR, 2013; Salarijazi et al., 2012; Afkhami et al., 2007), the most downstream gauging station but still located approximately 200 km upstream of the confluence with the SAR. Due to large scale water developments, the mean annual discharge of the Karun has experienced a consistent negative trend from 818 $m^3$/s to 615 $m^3$/s before and after 1963, respectively (UN-ESCWA and BGR, 2013). Whereas Salarijazi et al. (2012) reported a mean annual river discharge at Ahvaz of 1,442 $m^3$/s for the period 1954-2005, the mean monthly river discharge for the period between 1978 and 2009 was only 667 $m^3$/s (personal communication Dr. Meysam Salarijazi). However, the Karun river discharge into the SAR is believed to have decreased even more in recent years due to continued increases in water abstractions upstream.  The combination of tide and fluctuating river discharge makes it difficult to recognize the real extent of salt intrusion and its impact on the horizontal salinity pattern along the river under different conditions.

Adding to the complexity of studying salt intrusion in the Shatt al-Arab, is that it is the border river between Iraq and Iran, with strict security conditions. This does not make it easy to organize hydrometric surveys by speedboat and carry out salinity observations during an entire tidal cycle. As a result, the field data collected during this study and the results obtained by the analytical model form a unique data set for the region.

## 3  Theory of the analytical model

During a tidal cycle, the tidal velocity is near zero just before the tidal current changes direction. This situation is known as high water slack (HWS) just before the direction changes seaward, and low water slack (LWS) just before the direction changes landward. The model originally proposed by Savenije (1989), calibrated with measurements made at HWS, describes the salinity distribution in convergent estuaries as a function of the tide, river flow and geometry, using the Van der Burgh's coefficient ($K$) and the dispersion coefficient ($D_0$) at the mouth. A conceptual sketch of the 1-D model of salt intrusion is shown in Fig.3.

The geometry of an estuary can be presented by exponential functions describing the convergence of the cross-sectional area and width along the estuary as:





$A = A_o \exp^{-\frac{x}{a1}},$        for $0 < x \le x_1$      (1)
$A = A_1 \exp^{-\frac{(x-x1)}{a2}},$      for $x > x_1$      (2)
$B = B_o \exp^{-\frac{x}{b1}},$        for $0 < x \le x_1$      (3)
$B = B_1 \exp^{-\frac{(x-x1)}{b2}},$      for $x > x_1$      (4)
where $A_o$ and $B_o$ are the cross-sectional area $[L^2]$ and width $[L]$ at the estuary mouth ($x=0$), $A_1$ and $B_1$ are the cross-sectional area and width
at the inflection point ($x=x_1$), and $a_{1,2}$ and $b_{1,2}$ are the cross-sectional and width convergence lengths $[L]$ at $x \le x_1$ and $x > x_1$, respectively.
Combining (1) with (2) and (3) with (4) describes the longitudinal variation of the depth:
$h = h_o \exp^{-\frac{x(a1-b1)}{a1b1}},$      for $0 < x \le x_1$      (5)
$h = h_1 \exp^{-\frac{(x-x1)(a2-b2)}{a2b2}},$      for $x > x_1$      (6)
where $h$, $h_o$ and $h_1$ are the cross-sectional average water depths $[L]$ at distance $x$ from the mouth, at the estuary mouth, and at the inflection
point respectively.
Integrating the geometry equations into the salt balance equation of Van der Burgh (1972) yields a steady state longitudinal salinity
distribution along the estuary (see Savenije 2005) under HWS condition:
$S - S_f = S_o - S_f (\frac{D}{D_o})^{\frac{1}{K}},$      for $0 < x \le x_1$      (7)
$S - S_f = S_1 - S_f (\frac{D}{D_1})^{\frac{1}{K}},$      for $x > x_1$      (8)
where $D_o$, $D$, and $D_1$ $[L^2T^{-1}]$ are the dispersion coefficient at the estuary mouth, at any distance $x$, and at the inflection point, $S_0$, $S_1$ and $S$
$[ML^{-3}]$ are the salinity at the estuary mouth, inflection point, and distance $x$ respectively, $S_f$ is the fresh water salinity, and $K$ is the Van der
Burgh coefficient which according to Savenije (2005) has a value between 0 and 1; where

$\frac{D}{D_o} = 1 - \beta_o \left( \exp\left(\frac{x}{a_1}\right) - 1 \right),$      for $0 < x \le x_1$      (9)

and





$$\frac{D}{D_1} = 1 - \beta_1 \left( \exp\left(\frac{x - x_1}{a_2}\right) - 1 \right), \qquad \text{for } x > x_1 \qquad (10)$$

with

$$\beta_o = \frac{K a_1 Q_f}{D_o A_o}, \qquad\qquad \text{for } 0 < x \le x_1 \qquad (11)$$

$$\beta_1 = \frac{K a_2 Q_f}{D_1 A_1}, \qquad\qquad \text{for } x > x_1 \qquad (12)$$

$\beta_o$ and $\beta_1$ are the dispersion reduction rate [-] at the estuary mouth and at the inflection point, respectively, and $Q_f$ is the freshwater discharge.

The salt intrusion model is used to estimate the salt intrusion length, which can be determined using low water slack (LWS, the lower extreme salt intrusion), high water slack (HWS, the upper salt intrusion), or tidal average (TA, the average of full tidal cycle). Savenije (2012) proposed to calibrate the model on measurements carried out at HWS. This is to obtain the maximum salt intrusion over the tidal cycle. The salinity distribution can be computed at LWS and TA based on the relation between salinity distributions during the three conditions. The salt distribution curve at HWS could be shifted downstream over a horizontal distance equal to the tidal excursion length ($E$) and half of the tidal excursion length ($E/2$) to obtain the salt distribution curve at LWS and TA conditions respectively. The model variables can be determined from field observations and shape analysis; while the two parameters $K$ and $D_0$ remain unknown, in addition to $Q_f$, which is difficult to determine in the tidal region. To facilitate the calibration process, $D_0$ and $Q_f$ are combined in one variable, the mixing coefficient $\alpha_0$ [$L^{-1}$]:

$$\alpha_0 = \frac{D_0}{Q_f}, \qquad\qquad (13)$$

After model calibration, the values for $K$ and $\alpha_0$ are known and the salinity at any point along the estuary can be calculated. Finally the salt intrusion length ($L$) during HWS is obtained by:

$$L^{HWS} = x_1 + a_2 \ln\left(\frac{1}{\beta_1} + 1\right), \qquad\qquad (14)$$

The calibration parameters can be obtained based on field measurements, but to turn the model into a predictive model, a separate equation for $D_0$ is required. A predictive equation for $D_0$ was presented by Savenije (1993), and then improved by Gisen et al. (2015b), who moved the boundary condition to a more identifiable inflection point $x_1$, based on observations made for a large number of estuaries worldwide as:

$$D_1 = 0.1167 \, E_1 v_1 N_R^{0.57}, \qquad\qquad (15)$$





with

$$N_R = \frac{\Delta \rho \; gh \; Q_f T}{\rho \; AE \upsilon^2},$$ (16)

and

$$E = \frac{\upsilon T}{\pi},$$ (17)

$N_R$ is the estuarine Richardson number [-], the ratio of potential energy of the buoyant freshwater to the kinetic energy of the tide, $\rho$ and $\Delta \rho$ [ML$^{-3}$] are the water density and the density difference over the intrusion length, $g$ is the gravitational acceleration [LT-2], $T$ is the tidal period [T], $\upsilon$ is the velocity amplitude [LT-1], and $E$ is the tidal excursion [L].

This study tests the predictive performance of the 1-D analytical salt intrusion model, combined with new revised predictive equations to analyze the real extent of seawater intrusion in the SAR estuary under different river discharge conditions.

**4 Data collection**

The 1-D analytical salt intrusion model is based on a number of parameters that can be obtained through field surveys. Variables such as $K$ and $D_0$ are not directly measureable and therefore they are obtained by calibrating the simulated salinity curve to the datasets from the salt intrusion measurements. For this study four measurement campaigns were conducted, mainly measuring salt concentrations and water levels. The measurements took place during the wet and dry periods at spring and neap tides. These were on 26 March 2014 (neap-wet), 16 May 2014 (spring-dry), 24 September 2014 (spring-dry), and 5 January 2015 (spring-wet).

Salinity measurements were conducted at the moment just before the flow changes direction (HWS and LWS). The HWS and LWS represent the envelope of the vertical salinity variation during tidal cycles, and are also used to determine the longitudinal tidal excursion. A moving boat technique was used in the field survey in which the boat moved with the speed of the tidal wave to capture the slack moment. Starting from the mouth of the estuary and in the middle of the course, the salinity variations during the tidal cycle were observed. A conductivity meter, YSI EC300A (https://www.ysi.com) with a cable length of 10 m, was used to measure the vertical salinity profile for each meter depth from the bottom to the surface, and it was done repetitively at an interval of 3-4 km (longitudinally) until the river salinity was reached (in this case 1.5 kg/m$^3$).





The required information on river discharge and cross-sectional profile were provided by the local water authority. It is difficult to measure the discharge accurately in an estuary considering the tidal fluctuation. Hence, the discharge data from the nearest (most downstream) station was used in the analysis. The daily stream flow data of all the tributaries within the country were obtained from the Department of Water Resources in Iraq. However, there were no data on the discharge of one tributary, the Karun River, being located in neighboring Iran. Experts of the water resources authority in Basra indicated that average discharge of the Karun River was estimated at 40 m$^3$/s. River cross sections data were collected based on the last survey carried out in 2012 by GDSD (General Directorate of Study and Design).

# 5 Salinity modelling

## 5.1 Geometric characteristics

Results of the cross-sectional area, width, and depth are presented in a semi-logarithmic scale plot in Fig. 4. This Figure shows a good agreement between the computed cross-sectional areas $A$, width $B$, and depth $h$ based on Eqs. (1)-(6) and the observed data, except for the part between 40 and 50 km, which is shallower in comparison to the rest of the estuary. The cross-sectional area $A$ and width $B$ are divided into two reaches with the convergence length $a_1$ and $a_2$ of 22 km and 26 km respectively (see Table 1). The geometry changes in decreasing pattern landwards following an exponential function. In an alluvial estuary, the wide mouth and shorter convergence length on the seaward part is generally wave dominated, while the landward part with longer convergence length is tide dominated. The average depth $h$ is almost constant with a very slight decrease along the estuary axis (a depth convergence length of 525 km).

## 5.2 Vertical salinity profile

In Fig. 5 the results of the observed vertical salinities profile at HWS are presented. It can be seen that the salt intrusion mechanism is well mixed for the entire observation period. During the wet period when river discharge is relatively high, partially mixed condition can be observed particularly at the downstream area (Fig. 5a and 5d). In the neap-wet condition as shown in Figure 5a, there is more stratification and the partially mixed pattern occurs in almost the entire stretch of the estuary. This is because at neap tide, the tidal flows are small compared to the high fresh water discharge during the wet season. Conversely, during the spring-dry period when the river discharge is significantly low and the tidal range is large (Fig. 5b and 5c), the vertical salinity distribution along the estuary is well mixed.

## 5.3 Longitudinal salinity profile

The measurements of salinity during HWS and LWS are presented in Fig. 6. Calculations of the longitudinal salinity profiles are based on Eqs. (7)-(14), where the dispersion $D$ decreases over $x$ until it reaches zero at the end of the salt intrusion length. Coefficients $K$, $D_0$, and $E$



were calibrated to obtain the best fit between measured salinity data and simulated salinity variations. The longitudinal salinity distributions during a tidal cycle are demonstrated by three curves: (1) the maximum salinity curve at HWS; (2) the minimum salinity curve at LWS; (3) the average of HWS and LWS represent the average salinity curve at TA. Tidal excursion ($E$) is determined from the horizontal distance between the salinity curves of HWS and LWS. This distance is considered constant along the estuary axis during the tidal cycle. In this study, the tidal excursion is found to be 14 km on 24 September and 10 km for the other observations (Table 2).

The results show good agreement between measured and simulated salinity profiles with few deviations between the observed and modeled salinities. The small deviations may be due to the timing errors in which the boat movement speed did not coincide exactly with the tidal wave. In Fig. 6 (a and d), it can be seen that the measured salinity at distance 20 and 24 km during HWS are higher than the simulated values. There is a sub-district (with considerable agricultural communities) and a commercial harbor, and it is believed that all of their effluents and drainage water are discharged into the river. This could be the reason for the salinity to be a little higher than expected. In Fig. 6c, the last measurement point is lower than the simulated one. This may be due to the relatively shallow stretch between 40 and 50 km, which can substantially reduce the salt intrusion. Also a timing error may be an explanation for this deviation: the boat did not move fast enough as it was delayed for short stops at police checkpoints.

All the field surveys indicate that the maximum salinity at the mouth ranged from 24 to 35 kg/m$^3$ (Table 2). The lowest maximum salinity is during the neap-wet period and the highest is during the spring-dry period. It can be seen that the sea water intrudes furthest in September (spring-driest period) and shortest in March (neap-wet). These finding are logical because during wet season, the estuary is in a discharge dominated condition and the lower tide (neap) can be easily pushed back by the river discharge. On the other hand, during the dry season the estuary is tide dominated and the higher tide (spring) managed to travel further inland without much obstruction (low fresh water discharge). The tidal ranges recorded during field surveys are 1.7, 3.2, 2.1, and 2.6, respectively as same date shown in Fig. 6(a)-(d).

Besides sea water intrusion, human activities at the upstream part of the estuary also contribute to the salinity levels along the river. From observations, the river salinity at the inland part varies in space and time between 1-2 kg/m$^3$. Thus, the salt concentrations are the result of a combination of anthropogenic and marine sources (Abdullah et al., submitted). The findings from the longitudinal salinity distribution indicate that there is a need to analyze and classify the effects of natural and anthropogenic factors on estuary salinity.

**5.4 The predictive model**

The dispersion coefficient $D$ is not a physical parameter that can be measured directly. It represents the mixing of saline and fresh water, and can be defined as the spreading of a solute along an estuary induced by density gradient and tidal movement. Knowing the river discharge is crucial for determining a dispersion coefficient $D$ from Eq. (9). However, it is difficult to measure the river discharge accurately in the tidal region due to the tidal fluctuation. In this study, the river discharge data on the days of the measurements were used from the gauging station located at the most downstream part of the river network.





For the situation where measured salinity are known the dispersion coefficient $D_0$ and the salt intrusion length $L$ at HWS were calibrated by
fitting the simulated salinity curve (Eq. 7-14) against the field data. In case no field data are available, the dispersion coefficient $D_1$ was
estimated using Eq. 15. The predicted D1 then was used to determine the predicted $D_0$ (using Eq.9) and $L$ (using Eq.14). Comparisons
between the calibrated and predicted values were done to evaluate the performance of the model. The prediction performance was
evaluated with two model accuracy statistic: The root mean squared error ($E_{\mathrm{RMS}}$) and Nash-Sutcliffe Efficiency ($E_{\mathrm{NS}}$) (Equation 18and 19,
respectively).
$E_{\mathrm{RMS}} = \sqrt{\frac{1}{n} \sum_{i=1}^{n}(P_i - O_i)^2}$        (18)
$E_{\mathrm{NS}} = 1 - \frac{\sum_{i=1}^{N}(O_i - P_i)^2}{\sum_{i=1}^{N}(O_i - \bar{O})^2}$        (19)
where $P$ and $O$ are the predictive and observed variables, respectively, and $\bar{O}$ is the observed mean. The index ($E_{\mathrm{NS}}$) ranges from -∞ to1. It
describes the degree of accurate prediction. The efficiency of one indicates complete agreement between predicted and observed variables,
whereas efficiency less than zero indicate that the prediction variance is larger than the data variance.
Figure 7 presents poor correlations between the calibrated and predicted values of $D$, the situation is better in case of $L$ values. Table 4
displays the correlation between predicted and measured values. The $E_{\mathrm{NS}}$ obtained for $D$ is -0.09 reflects week predictive performance.
Generally the model appears to overestimate the values of the dispersion coefficient compared to the calibrated ones during the wet period
and to underestimate the value during the drought period in September. This could be due to the use of the measured discharge at the end of
the tidal domain, which gives higher or lower values than the exact fresh water discharging into the estuary, as it does not account for the
discharge of the Karun River at the downstream end and the water consumptions and water losses within the system (see Table 3). The
SAR is the main freshwater source for irrigation, domestic and industrial activities in the region. Hence, water consumptions could highly
affect the performance of a predictive model especially in the region where water withdrawals can considerably reduce river discharge into
the estuary.



In order to reduce the uncertainty in the discharge data, some alternative approach has to be adopted. Gisen et al. (2015a) estimated the
discharges for the downstream areas by extrapolating the correlation of the gauged area to the ungauged areas. Cai et al. (2014) developed
an analytical approach to predict the river discharge into an estuary based on tidal water level observations. This method is only applicable
in estuaries with a considerable river discharge compared to the tidal flows. In this study, a simple approach has been used to assess the
discharge in the SAR estuary by deducting the water withdrawals in the downstream region from the discharge data collected at the lowest
gauging points. In a similar way also the average discharge of the Karun was estimated (Table 3). Data on water withdrawals were
collected from the water resources authority and water distribution departments. Besides irrigation and domestic supply, the industrial


sector, including the oil industry, is also a significant water user. Unfortunately this study could not obtain information on water usage and disposal by the oil industry.

The adjusted river discharge data are then applied in the predictive model to evaluate the improvement of these changes in predicting values of $D$ and $L$. The results obtained after the adjustment are shown in Fig. 8. The figures demonstrate the improvements in predicting the dispersion and maximum salt intrusion length and show the importance of computing the fresh water discharge accurately. Furthermore the correlations between predictive and observed values are improved for both $D_0$ and $L$, 0.46 and 0.9 respectively, the $E_{\mathrm{RMS}}$ also reduced to 60 m$^2$/s and 4 km for $D_0$ and $L$, respectively (Table 4).

The prediction performance of the model is demonstrated in Fig. 9, where the salinity curves were computed from the predictive equation of $D_1$ and the adjusted river discharges. Figure 9 shows that the prediction salinity curves perform very well compared to the calibrated one during all periods, except January 2015. This could be attributed by the average discharge used for the Karun River, in which the value is lower than the actual discharge, being the month of January, is in the mid of the wet season. At such time the SAR is expected to receive high flow rates from the Karun River. On the other hand, during this season more return flows are drained into the SAR from the large irrigation scheme serviced by the Karun water system, increasing anthropogenic salinity levels. Accurate estimation of river discharge into the estuary is important in improving the predictive skill of the model.

The ultimate objective of the modeling is to assess the influence of upstream development on the estuarine environment, and also to find the real extent of salt intrusion. The salinity distribution along the estuary is highly linked to upstream conditions, such as flow regulation and water withdrawals. For the purpose of improving the SAR estuary management, the model can lead to estimate a salt intrusion length for a given fresh water discharge. This is useful for water supply managers to determine the appropriate location (salinity free region) for water intake stations. Fig. 10 demonstrates the salt intrusion length ($L$) associated with different river discharges corresponding to water released from the Tigris River into the SAR. The salt intrusion lengths are plotted against a range of fresh water discharge from 5 to 120 m$^3$/s. The main finding is that the length of salt intrusion increases in a non-linear way with decreasing river discharge. The salt intrusion length is very sensitive for river discharge when the flow is low. From the plot it can also be seen that the maximum salt intrusion could reach 92 km from the SAR estuary at 5 m$^3$/s river discharge. This outcome exceeds a preliminary estimate by Abdullah et al. (submitted) based on a one-year data series, who found the salinity to reach up to 80 km considering the annual salinity peaks along the river. An 80 km intrusion length corresponds with a measured river discharge of 58 m$^3$/s, whereas for the predictive model this distance corresponds to a much lower discharge (7 m$^3$/s). It is, however likely that the true river discharge was lower, since during the lowest discharge the irrigation demand is relatively high. It should also be realized that in the region of 40-50 km the depth and cross-sectional area are substantially less. Such a shallow reach can reduce the salt intrusion length substantially, as can be seen from Eq. (14), where $\beta_1$ is inversely proportional to $A$.





## 6 Conclusion

A one-dimensional analytical salt intrusion model was applied to the SAR estuary based on four survey campaigns in 2014 and 2015. This model is used to determine longitudinal salinity distribution and the length of salt intrusion. The analytical model is shown to describe well the exponential shape of the estuary in the upstream direction. Moreover, the results show good agreement between computed and observed salinity under different river conditions. This indicates that the analytical model is capable of describing the extent of seawater intrusion along the SAR estuary.

Results for the dispersion coefficient $D_o$ indicate that the measured river discharge out of the tidal range is higher than the real discharge into the estuary. This can be attributed to water withdrawals along the tidal domain. In case of low river discharge, water withdrawals have a considerable effect on the predicted salt intrusion length. The river discharge into the estuary was revised considering water withdrawals of irrigation and domestic sectors. Using adjusted river discharge improved performance of the predictive equations. For further improvement, it is recommended to obtain more accurate estimation of the river discharge into the estuary.

Seawater intrusion is driven by the discharge kinetics from tidal seawater and the hydrostatic potential energy from fresh water fluctuations. Intrusion lengths of 38, 40, 65, and 43 km correspond with tidal ranges of 1.7, 3.2, 2.1, and 2.6 m during March 2014, May 2014, September 2014, and January 2015, respectively. The longer salt intrusion distance is caused by low river discharge, as evident for September (dry period).

The SAR is the main source of freshwater for daily consumption and irrigation. Decreased freshwater discharge and increased seawater intrusion will exacerbate an already critical situation in that important agricultural and ecological region. The model shows a scenario in which decreasing river discharge, considered a likely event, can result in an increase in seawater intrusion further upstream to a distance of 92 km. Additional salinity sources from anthropogenic activities will diminish the volume of fresh water leading to very serious health problems, water and food insecurity. Calibration of the model can be enhanced with further monitoring of discharge and salinity from all the tributaries and used to make new estimates of longitudinal salinity distribution under extreme conditions. Preventing salt intrusion of these magnitudes can only be achieved if the water quantity and quality of the upstream sources as well as along the SAR are promptly and strictly regulated

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

**Table 1**. The geometry characteristic of the SAR estuary

| $A_0$ (m$^2$) | $A_1$ (m$^2$) | $B_0$ (m) | $B_1$ (m) | $a_1$ (m) | $a_2$ (m) | $b_1$ (m) | $b_2$ (m) | $\bar{h}$ (m) |
|---|---|---|---|---|---|---|---|---|
| 8,050 | 4,260 | 910 | 531 | 22,000 | 160,000 | 26,000 | 230,000 | 7.9 |

Notes: $A_0$ and $A_1$ are cross-sectional areas at the mouth and inflection point respectively. $B_0$ and $B_1$ are channel widths at the mouth and inflection point respectively,
and $a_1$, $a_2$, and $b_1$, $b_2$ are locations of the convergence length of the cross-sectional area and width respectively. $\bar{h}$ is the average depth over the estuary length (of 60
km).
**Table2**. Characteristic values of the estuary including the maximum salinity at the mouth $S_o$, the river discharge $Q_f$, Tidal excursion $E$, Van
der Burgh coefficient $K$, the dispersion coefficient $D_o$, mixing number $\alpha_o$, and salt intrusion length $L$.

| Period | $S_o$ | $Q_f$ | $E$ | $K$ | $D_o$ | $\alpha_o$ | $L$ |
|---|---|---|---|---|---|---|---|
|  | (kg/m$^3$) | (m$^3$/s) | (km) |  | (m$^2$/s) | (m$^{-1}$) | (km) |
| 26 March 2014 | 24 | 109 | 10 | 0.65 | 403 | 3.7 | 32 |
| 16 May 2014 | 28 | 91 | 10 | 0.65 | 473 | 5.2 | 42 |
| 24 September 2014 | 34.6 | 48 | 15.5 | 0.65 | 442 | 9.2 | 65 |
| 05 January 2015 | 28 | 53 | 10 | 0.65 | 281 | 5.3 | 42 |

**Table 3**. Measured and adjusted river discharge considering water consumptions on the days of measurements.

| Date | Measured river discharge (not counting water abstractions and excluding the Karun inflows) (m$^3$/s) | Adjusted river discharge (deducting water abstractions and including the Karun inflows) (m$^3$/s) |
|---|---|---|
| 26 March 2014 | 109 | 114 |
| 16 May 2014 | 91 | 96 |
| 24 September 2014 | 48 | 58 |
| 05 January 2015 | 53 | 63 |





**Table 4**. Results of the model performance in terms of root mean squared error ($E_{RMS}$) and Nash-Sutcliffe efficiency ($E_{NS}$).

| Date | NSE | RMSE |
|------|-----|------|
| Measured river discharge | | |
| $D_0$ | -0.09 | 76 m$^2$/s |
| L | 0.75 | 6 km |
| Adjusted river discharge | | |
| $D_0$ | 0.46 | 60 m$^2$/s |
| L | 0.9 | 4 km |


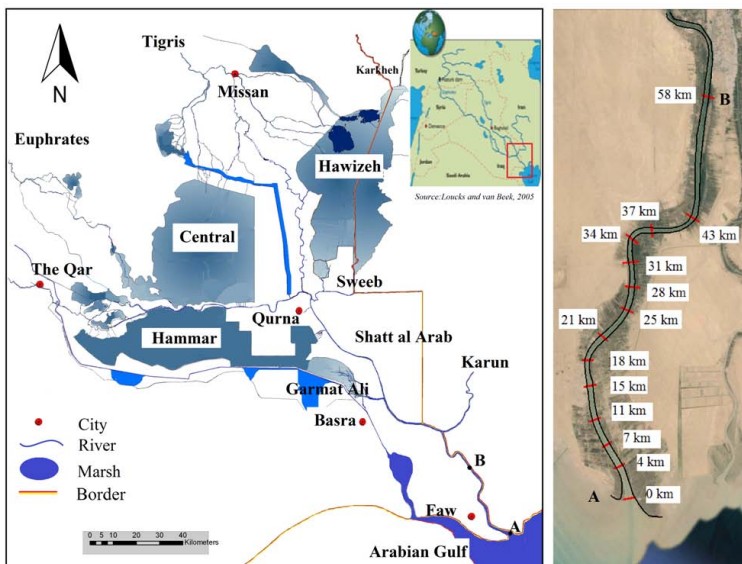

**Figure 1.** The Salient features of the Shatt al Arab region (left), the aerial view of the estuary from the Google earth with the measurement
locations (not to scale) (right).



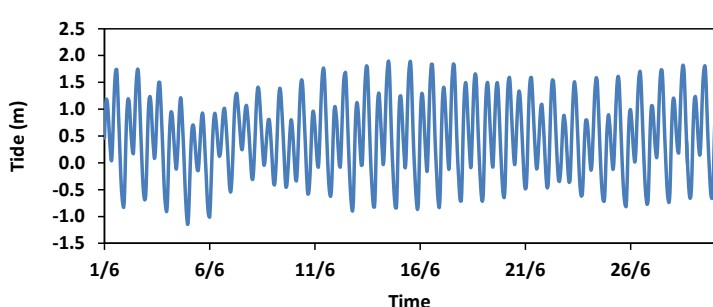

**Figure 2.** Tidal elevation at Faw station in June 2014.

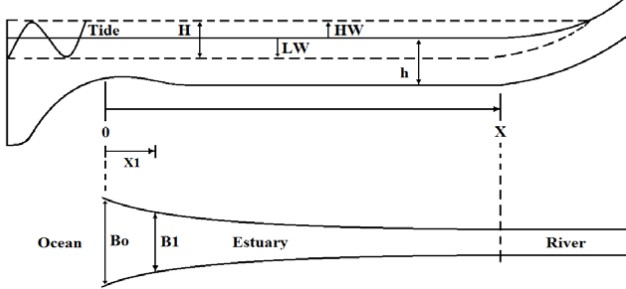

**Figure 3.** Sketch of the estuary the longitudinal profile and top view.





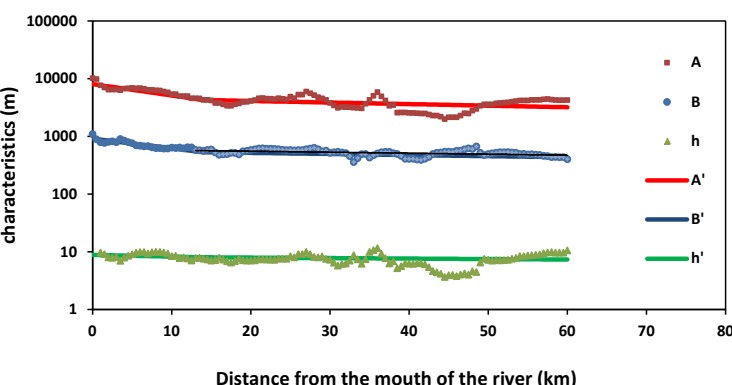

2
3 **Figure 4.** SAR geometric characteristics (A, B, h measured; A', B', h' equations 1-6).
4




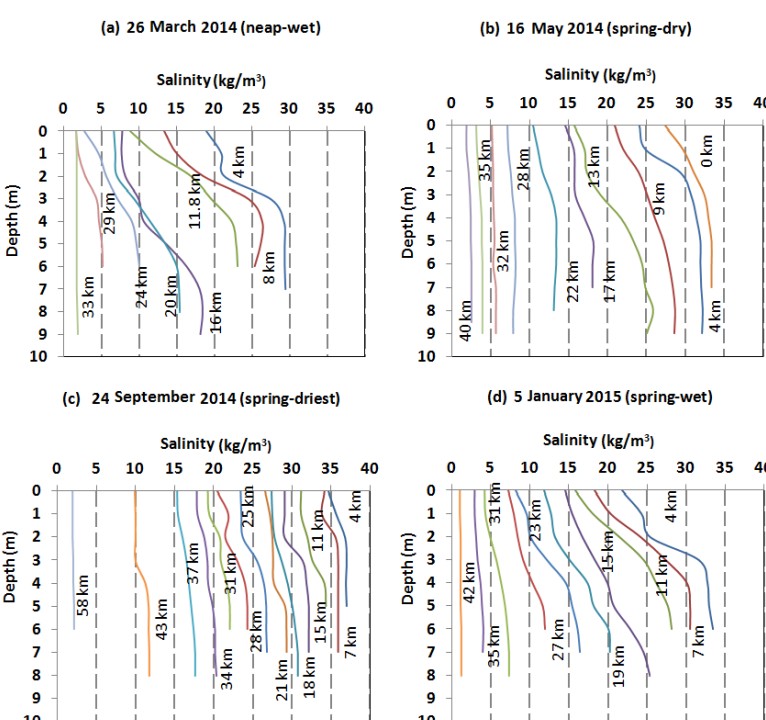

**Figure 5.** Vertical salinity distribution of the estuary measured between 0 and 58 km at HWS.





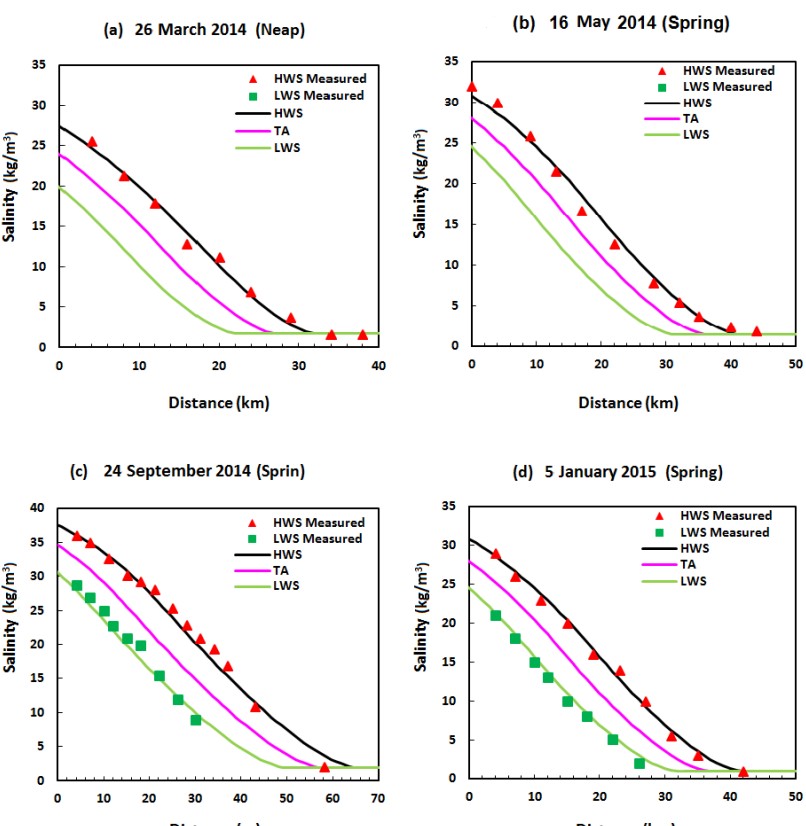

**Figure 6.** Predicted and measured salinity distribution during HWS, TA, and LWS.

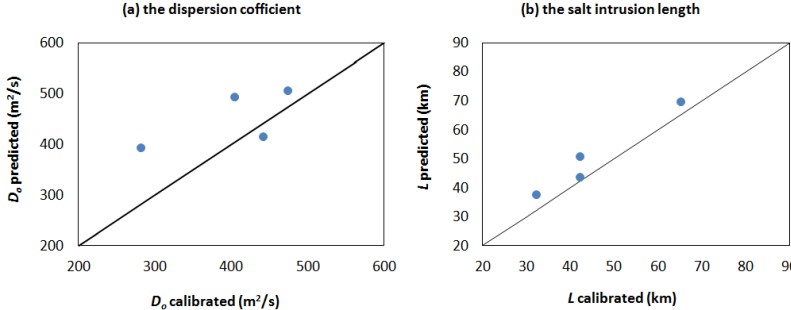

2 **Figure 7.** Comparison between the predicted and calibrated values of $D_0$ and $L$.

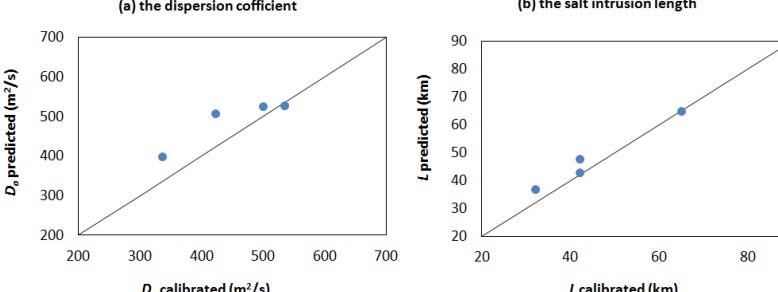

5 **Figure 8.** Comparison between the predicted and calibrated values of $D_0$ and $L$ using the improved discharge data.





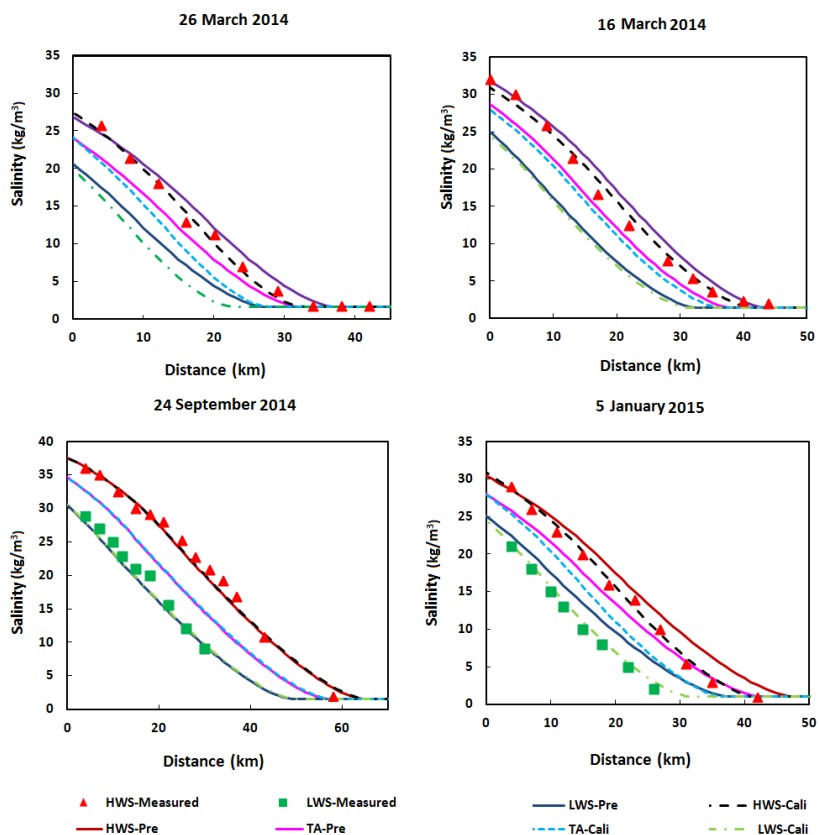

2    **Figure 9.** Compare the salinity curves of the calibrated results (dashed lines) and the predicted results (solid lines) to the observed salinity
3         during the four periods of the 2014.





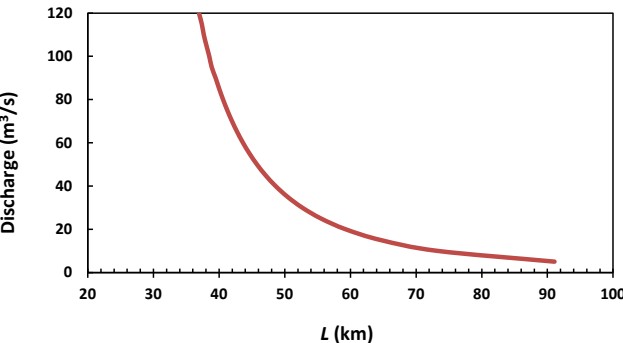

**Figure 10.** Relationship between river discharge and predicted salt intrusion length.