# Peer review of "Predicting the salt water intrusion in the Shatt al Arab estuary using an analytical approach"

_Hydrology and Earth System Sciences, 2016_

## Referee Comment (RC1) · Anonymous Referee #1 · 30 May 2016

The authors constructed a predictive model by analytical approach to predict the length of salt water intrusion in the Shatt al-Arab River estuary. Comparison with in-situ observation shows that model prediction would be improved by considering and estimating water abstractions through anthropogenic usage along the estuary. The overall presentation of the MS is clear, while revisions are needed to improve the scientific value of the MS, before it could be accepted by HESS.

Specifics:

1. In section 3 Theory of the analytical model, the analytical model equations are not newly derived by this MS, thus the derivation process should be put in the Appendix. The applications of the analytical model and new modifications to the equations should be clearly presented in section 3.

2. Results show that the seawater intrusion is controlled by the fresh water discharge and tide, how about other physical factors, such as the variation of sea level, wind direction, width and depth of the estuary? Considering adding discussion of these factors might improve the understanding to the scientific issue on salt water intrusion.

3. The applicability of the predictive model. Considering adding discussion on the model could be used to predict the salt water intrusion in what kind of estuary (wide, narrow, stratified or vertically mixed)?

4. Some sentence and words need to be modified. For example, in page 2 line 27, "Lui et al. (2004)" should be "Liu et al. (2004)"; in page 3 line 11, "This is by applying . . ." should be changed into " This is carried out by . . ."; in page 5 line 9, "combing (1) with (2), and (3) with (4) describes the . . .", seems should be "(1) with (3), and (2) with (4)" , should not simply use "combing", but use more specific words like " multiply, subtract, etc." in deriving the equations.

---

## Author Comment (AC1) · 11 Jun 2016

Interactive comment on "Predicting the salt water intrusion in the Shatt al Arab estuary using an analytical approach" By A. D. Abdullah et al.

Anonymous Referee #1

The authors constructed a predictive model by analytical approach to predict the length of salt water intrusion in the Shatt al-Arab River estuary. Comparison with in-situ observation shows that model prediction would be improved by considering and estimating water abstractions through anthropogenic usage along the estuary. The overall presentation of the MS is clear, while revisions are needed to improve the scientific value of the MS, before it could be accepted by HESS.

[Figure]

Our reply: Thank you very much for these constructive remarks. Below we shall reply to your comments, which we shall take into account in the revised manuscript.

Specifics: 1. In section 3 Theory of the analytical model, the analytical model equations are not newly derived by this MS, thus the derivation process should be put in the Appendix. The applications of the analytical model and new modifications to the equations should be clearly presented in section 3.

Reply: In section3 of the MS, the authors describe the equations used by the analytical model. Detailed description of the deduction of these equations is provided as reference through cited literature. The introduction of these equations was done along the definition of the basic terms, parameters, and description of the relation between the dispersion coefficient and river discharge. Authors consider that such type of information within the MS text helps the reader to better understand the presented content. Authors have no problem in changing this section if requested in the final stage of the MS editing.

2. Results show that the seawater intrusion is controlled by the fresh water discharge and tide, how about other physical factors, such as the variation of sea level, wind direction, width and depth of the estuary? Considering adding discussion of these factors might improve the understanding to the scientific issue on salt water intrusion.

Reply: The model provides a set of analytical equations that can describe salt intrusion in alluvial estuary and predict the impact of interventions such as river discharge, estuary dredging and sea level rise. Wind is not a factor in these equations.

3. The applicability of the predictive model. Considering adding discussion on the model could be used to predict the salt water intrusion in what kind of estuary (wide, narrow, stratified or vertically mixed)?

Reply: The applicability of the model is limited to partially and well-mixed in funnel-shaped alluvial estuaries, where the amount of fresh river inflow per tidal period is small

in relation to the tidal prism (the quantity of ocean water which enters into the estuary during the flood current). The authors will include this remark in the final version of the MS.

4. Some sentence and words need to be modified. For example, in page 2 line 27, "Lui et al. (2004)" should be "Liu et al. (2004)"; in page 3 line 11, "This is by applying... " should be changed into " This is carried out by ... "; in page 5 line 9, "combing (1) with (2), and (3) with (4) describes the ... ", seems should be "(1) with (3), and (2) with (4)" , should not simply use "combing", but use more specific words like " multiply, subtract, etc." in deriving the equations.

Reply: Thank you for these suggestions. All specified corrections will be carried out while preparing the revised MS.

---

## Referee Comment (RC2) · Anonymous Referee #2 · 12 Sep 2016

The authors presented an interesting application of a 1D analytical salt water intrusion model on the SAR system. The novelty of the study situates in 2 aspects. First, new field data on the salt water intrusion problem on the SAR are collected, presented and analysed, which is as such already a good scientific achievement, knowing the problem of data collection in this difficult geopolitical environment. Second, the analytical model intrusion model was augmented with the predictive equation for the dispersion coefficient of Gisen et al (2015). This model combination shows acceptable performance, conditional to a good estimate of discharge data in the SAR system.

The manuscript is well presented, clearly written, concise and well formatted. Some small editorials are given in the annotated manuscript, but 2 points of concerns should still be addressed in a small minor revision.

[Figure]

1. In some places in the manuscript, authors refer to personal communications and estimates that are made by local water authority on discharges in part of the SAR system (in particular the Karun tributary). The support for these estimates seems very poor. It would be appropriate to try to consolidate these statements, eventually by clarifying what quantitative information was used by local experts to make such assessments.

2. The authors focus on the impact of Qf on D estimates in the combined approach. Yet, D is also affected by T, E, v. It would be good to perform a sensitivity analysis confirming that Qf is indeed the driving factor in the D uncertainty. Also, the quality of the calibration is not ver well presented. From Table 2, we can not infer the precision of the calibrated D values.

Please also note the supplement to this comment:
http://www.hydrol-earth-syst-sci-discuss.net/hess-2016-141/hess-2016-141-RC2-supplement.pdf

[Figure]

**Supplement:**

[revised manuscript text omitted]

---

## Author Comment (AC2) · 21 Sep 2016

The authors presented an interesting application of a 1D analytical salt water intrusion model on the SAR system. The novelty of the study situates in 2 aspects. First, new field data on the salt water intrusion problem on the SAR are collected, presented and analysed, which is as such already a good scientific achievement, knowing the problem of data collection in this difficult geopolitical environment. Second, the analytical model intrusion model was augmented with the predictive equation for the dispersion coefficient of Gisen et al (2015). This model combination shows acceptable performance, conditional to a good estimate of discharge data in the SAR system.

The manuscript is well presented, clearly written, concise and well formatted. Some small editorials are given in the annotated manuscript, but 2 points of concerns should

still be addressed in a small minor revision.

Our reply: Thank you very much for these constructive remarks. Below we shall reply to your comments, which we shall take into account in the revised manuscript.

Specifics: 1. In some places in the manuscript, authors refer to personal communications and estimates that are made by local water authority on discharges in part of the SAR system (in particular the Karun tributary). The support for these estimates seems very poor. It would be appropriate to try to consolidate these statements, eventually by clarifying what quantitative information was used by local experts to make such assessments.

Reply: Dr. Salarijazi, our contact source, is a researcher at the Department of Hydrology, Hydrology and Water Resources, Faculty of Water Sciences Engineering, Shahid Chamran University of Ahvaz, Khuzestan, Iran) is a researcher and part of his work is about the hydrology of the Karun River. Based on the contact with him, he provided the first author valuable information and data including tidal level at 10 minutes intervals at khorramshahr and daily river discharges at Ahvaz for the period 1978-2009.

The experts information are based on several measurements of the water levels and river discharges. Measurements were made during the entire tidal cycle, upstream and downstream the point of discharge from the Karun River into the SAR. This is done just to roughly estimate the discharge of the Karun.

2. The authors focus on the impact of $Q_f$ on D estimates in the combined approach. Yet, D is also affected by T, E, v. It would be good to perform a sensitivity analysis confirming that $Q_f$ is indeed the driving factor in the D uncertainty. Also, the quality of the calibration is not very well presented. From Table 2, we cannot infer the precision of the calibrated D values.

Reply: the other parameter such as T (tidal period), E (tidal excursion), and v (tidal velocity amplitude) have been considered constant along the estuary axis. This was

shown by Savenije (2005) to be a valid assumption with the saline area of an estuary. In fact these parameters have only one unknown variable, which is the tidal velocity. The tidal period T is given (semi-diurnal tide). The relation between the parameters is E=(vT)/pi. E can be observed from salinity observations. The difference between the HWS and LWS salinity curve determines E and hence v. Thus E and v have been estimated on the basis of the salinity survey (see Figure 6). The dispersion depends on the river discharge and the observed tidal excursion. The tidal excursion can be easily observed. So, in this study the focus is on the two unknown parameters, D and Q. A new table will be added to show the calibrated D values. .

Please also note the supplement to this comment: http://www.hydrol-earth-syst-sci-discuss.net/hess-2016-141/hess-2016-141-RC2- supplement.pdf Reply: Thank you for all comments. All specified corrections will be carried out while preparing the revised MS.